# Losartan Alleviates the Side Effects and Maintains the Anticancer Activity of Axitinib

**DOI:** 10.3390/molecules27092764

**Published:** 2022-04-26

**Authors:** Ying Fu, Rengui Saxu, Kadir Ahmad Ridwan, Jiaping Yao, Xiaoxuan Chen, Xueping Xu, Weida Zheng, Peng Yu, Yuou Teng

**Affiliations:** 1China International Science and Technology Cooperation Base of Food Nutrition/Safety and Medicinal Chemistry, College of Bioengineering, Tianjin University of Science and Technology, Tianjin 300457, China; fuyingying605@163.com (Y.F.); 17320056317@163.com (R.S.); ahmad.ridwan.kadir@gmail.com (K.A.R.); yaojiaping128@163.com (J.Y.); cxx8191@163.com (X.C.); dlxuxueping@163.com (X.X.); 2Medical College, Yanbian University, No. 977 Gongyuan Road, Yanji 133002, China; a1679709514@163.com

**Keywords:** axitinib, VEGF receptor inhibitor, losartan, angiotensin receptor blockers, side effect, antitumor

## Abstract

Axitinib is one of the most potent inhibitors of the vascular endothelial growth factor (VEGF) receptor and shows strong antitumor activity toward various malignant tumors. However, its severe side effects affect the quality of life and prognosis of patients. Losartan, which functions as a typical angiotensin receptor blocker, controls the average arterial pressure of patients with essential hypertension and protects against hypertension-related secondary diseases, including proteinuria and cardiovascular injury. To explore the effects of losartan on side effects caused by axitinib and its antitumor activity, several animal experiments were conducted. This study first analyzed and explored the effect of losartan on the amelioration of side effects in Wistar rats caused by axitinib. The results showed that the systolic blood pressure of Wistar rats was significantly increased by about 30 mmHg in 7 days of axitinib treatment, while the combination of losartan significantly reduced the blood pressure rise caused by axitinib. The Miles experimental model and mouse xenograft tumor model were further used to evaluate the effect of losartan on the antitumor effect of axitinib. The result clearly demonstrated that losartan has no significant influence on axitinib-related low vascular permeability and antitumor activity. In summary, our results showed that the combination of axitinib and losartan significantly reduced the side effects and maintained the antitumor effects of axitinib. This study provides information for overcoming VEGF receptor inhibitor-related side effects.

## 1. Introduction

Axitinib [1], a secondary drug for advanced renal carcinoma, was used in a double-blind clinical test conducted for patients who had previously received standard antitumor treatments [2]. The results of axitinib significantly prolonged the progression-free survival [3] when contrasted with sorafenib. It has also been shown to have good therapeutic effects on other malignant tumors such as lung cancer [4], colorectal cancer [5], breast cancer [6], melanoma [7], and thyroid cancer [8]. As a broad-spectrum vascular endothelial growth factor (VEGF) receptor inhibitor, IC_50_ values of axitinib were 0.1, 0.2, and 0.1–0.3 nM for VEGF receptors 1–3, respectively [1]. Although axitinib has a potent antitumor effect, it is often accompanied by side effects [9] including diarrhea, hypertension, fatigue, proteinuria, abnormal liver function, hypothyroidism, etc. Studies have pointed out that the side effects caused by axitinib are mainly related to its blocking effect on vascular endothelial growth factor receptors [10]. The detection and domination of medication-related side effects [11] are the prognostic key points of clinical application. It is momentous for the optimization of the medication regimen of axitinib to improve the quality of life of patients.

Losartan, as a typical angiotensin receptor blocker [12], not only controls the average arterial pressure [13] in patients with essential hypertension, but also has a protective effect on hypertension-related secondary diseases [14] including proteinuria and cardiovascular damage. Clinically, it is mainly used for patients with organic diseases [15] such as hypertension accompanied by myocardial infarction, left ventricular hypertrophy, etc. At the same time, losartan is well-tolerated and has fewer side effects related to medication [16]. In addition to its direct antihypertensive effect, losartan plays an important role in repairing myocardial hypertrophy, vascular remodeling, and other cardiovascular structural changes, as well as stroke, diabetes, and end-stage renal disease. These effects are mainly related to its protective effect on endothelial cells [17]. More and more studies about losartan have been focused on its applications in antitumor therapy for some time [18]. The protective ability of losartan [19] on hypertension-related diseases and microvascular disease may also be beneficial to the control and treatment of side effects caused by axitinib. The protective effect of losartan on hypertension-related diseases and microangiopathy may also be beneficial to the control and treatment of VEGFR inhibitor-related side effects. Therefore, the study of the functions of losartan on the various side effects related to axitinib has considerable clinical significance.

In this project, in order to suppress the axitinib-related side effects, an axitinib–losartan combination strategy was launched. In the carefully designated experiments, we evaluated the antitumor effect and drug-related side effects of losartan when preinjected with axitinib. Then, we evaluated the antitumor effect and drug-related side effects of losartan when injected with axitinib. Firstly, the effect of axitinib on the blood pressure of Wistar rats was measured, and the effect of losartan on the blood pressure increase caused by axitinib was evaluated. In addition, the effects of axitinib combined with losartan on liver and kidney function indexes in Wistar rats were evaluated. Subsequently, the effects of the combined drugs on vascular endothelial function indicators were evaluated. To evaluate the effect of the combined drugs on skin vascular permeability, a mouse model of Miles assay was established. Losartan had no significant influence on axitinib-related low vascular permeability, and losartan caused a noteworthy increase of vascular permeability in this model. Finally, a mouse xenograft model was established to evaluate the inhibitory effect of axitinib, losartan, and their combination on renal cell carcinoma. The research provided a basis for the control of the side effects of targeted VEGF receptor inhibitors, and provided a reference for the new usage of angiotensin receptor blocker antihypertensive drugs.

## 2. Results

### 2.1. Effects of the Combination of Axitinib and Losartan on Systolic Blood Pressure

In order to evaluate the effect of axitinib on the systolic blood pressure of rats and the effect of losartan on axitinib-related hypertension and heart rate changes, healthy Wistar rats were given axitinib and losartan for 7 days.

The changes in systolic blood pressure and heart rate of the Wistar rats from 2 days before administration to 7 days of continuous administration are shown in Figure 1. It can be seen from Figure 1B that the heart rate of the axitinib group was at the highest level in each group, and the heart rate of the control group was the lowest. As shown in Figure 1A, the values of systolic blood were observably reduced in the losartan group and the combination group significantly, whereas the value of the axitinib group fluctuated remarkably, owing to the side effects caused by axitinib. On the 7th day of the experiment, the blood pressure values of each group had a significant difference, as Figure 1C shows. The increase in systolic blood pressure of rats after the administration of axitinib reached about 30 mmHg, and the combination group significantly reduced the increase in blood pressure caused by axitinib, with a systolic blood pressure of less than 10 mmHg contrasted with the axitinib group. Compared with the control group, the systolic blood pressure of the single-agent losartan group decreased slightly. In the series of trials, the experimental results showed that the administration of axitinib increased the systolic blood pressure in rats, while the combined antihypertensive drug losartan inhibited this medication-related side effect significantly. The above experiments proved the potential of losartan in improving the side effects caused by axitinib. After which, we continued to explore the effect of the combination group on lessening other indicators of side effects.

### 2.2. Effects of Axitinib and Losartan Combined Group on Liver and Kidney Function

Alanine aminotransferase (ALT) and aspartate aminotransferase (AST) are markers for liver function, which can sensitively reflect whether liver cells were damaged or not and the degree of damage. The effects of the two drugs and their combination on the levels of AST/ALT in serum and liver homogenate of rats were tested for the sake of evaluation of the effects of axitinib and losartan and their combination on liver function. Axitinib significantly increased AST level in the serum of rats by 2.17 times compared with the control group, while the combination of axitinib and losartan reduced the content of AST caused by axitinib up to 1.86 times (Figure 2A). The level of ALT in the serum is shown in Figure 2B, axitinib plus losartan reduced ALT level compared with the axitinib group (*p* < 0.05). Therefore, losartan can alleviate axitinib-related liver dysfunction by reducing AST and ALT levels in the serum.

So as to further evaluate the effects of axitinib and losartan as single agents and their combination on the glomerular filtration function in rats, the effects of the drugs on the levels of serum creatinine and urea nitrogen in rats were detected. Losartan significantly reduced the level of creatinine compared to axitinib (*p* < 0.01). The creatinine value of the axitinib group (430 μmol/L) was higher than control group (238 μmol/L), and the combination of losartan (312 mol/L) reduced the increase of creatinine caused by axitinib significantly (*p* < 0.05), which showed that losartan had a protective effect on the increase in creatinine levels caused by axitinib (Figure 2C). In the study of the urea nitrogen content in serum, it was found that axitinib and losartan alone and their combined drugs had no significant effect on the level of urea nitrogen (Figure 2D). The insignificant change in urea nitrogen was presumed to be due to the short duration of action of the drug on the rats, which led to the early stage of renal failure in the rats. The kit cannot detect the change of urea nitrogen when the glomerular filtration rate is less 50% than a normal rat. On the other hand, proteinuria as an important evaluation index of renal damage, urinary protein in the axitinib group was 1928 mg/L higher than the control group at 1565 mg/L, a significant increase (*p* < 0.05). The amount of urine protein in the combined group was 1520 mg/L, a significant reduction in the increase in urine protein caused by axitinib (*p* < 0.05), as shown in Figure 2E. Based on the above data, losartan can ameliorate the renal function damage related to axitinib.

### 2.3. The Effect of Combined Group on Microvascular Function

#### 2.3.1. Effects of Axitinib and Losartan Combined Group on Endothelin-1 and Endothelial Nitric Oxide Synthase (eNOs) in Serum

The effect of axitinib and axitinib plus losartan on endothelial function was evaluated by assessing the levels of endothelin-1 and eNOs in rat serum. As shown in Figure 3A, axitinib treatment significantly increased endothelin-1 levels in the serum compared to the control group (*p* < 0.05). After combined treatment with losartan, endothelin-1 in the serum decreased from 155.07 μmol/L to 98.51 μmol/L. Simultaneously, losartan alone had no significant effect on endothelin-1 levels in the rat serum compared to the control group. Endothelin-1 plays an important role in the state of vascular endothelial contraction and relaxation. It may also be one of the pathogenesis of VEGF receptor inhibitor-related hypertension. The reduction of endothelin-1 levels in the combination group suggested its improvement in reducing axitinib-related side effects. Figure 3B shows the eNOs levels in the serum. In the control group, the eNOs level was 194 μmol/L, which was higher than that in other groups. The eNOs levels in the axitinib group and losartan group were 165 and 168 μmol/L, respectively, which were significantly lower than that of the control group (*p* < 0.05). The level of eNOs in the axitinib plus losartan group was 141 μmol/L, which was the lowest value among all groups. eNOs is of great significance in the occurrence of side effects related to certain VEGF receptor inhibitors. The index evaluation results of eNOs also further confirmed the amelioration of side effects related to axitinib by losartan.

#### 2.3.2. Effects of Axitinib and Losartan as a Single Agent and Their Combination on Skin Vascular Permeability

Vascular permeability plays an important role in tumor metastasis and cancer progress and the VEGF receptor inhibitor axitinib inhibits VEGF-induced skin vascular permeability. A Miles assay was performed to assess the effect of axitinib and axitinib plus losartan on VEGF-A-induced skin vascular hyperpermeability. Each group was separated into VEGF-A group and phosphate-buffered saline (PBS) group. As shown in Figure 4, in the model group, VEGF-A remarkably increased the skin vascular permeability compared with PBS (170% vs. 100%), illustrating successful model establishment. Compared with the model group, axitinib treatment significantly reduced VEGF-A-induced skin vascular hyperpermeability (*p* < 0.001), whereas no changes were observed in the PBS group. In contrast, axitinib plus losartan showed similar effects as axitinib and significantly reduced VEGF-A-induced skin vascular hyperpermeability, demonstrating that losartan did not affect the vascular permeability-reducing effect of axitinib. Additionally, losartan alone did not alter VEGF-A-related vascular hyperpermeability but increased vascular permeability in the PBS group compared to that in the control group (*p* < 0.001). Axitinib reduced the vascular hyperpermeability related to VEGF-A, which may be one of the antitumor mechanisms of axitinib, and the combination of losartan can mediate the vascular low permeability caused by axitinib. It can be seen that losartan can inhibit the occurrence mechanism of axitinib-related side effects, and at the same time had no significant effect on its antitumor effect in inhibiting vascular permeability.

### 2.4. Effect of Axitinib and Axitinib Plus Losartan on the Xenograft Tumor Model

#### 2.4.1. Effect of Axitinib and Axitinib Plus Losartan on Xenograft Tumor Growth

The effect of axitinib and axitinib plus losartan on xenograft tumor growth was represented by tumor growth inhibition (TGI) ratio. As shown in Table 1, the TGIs following low- and high-dose axitinib treatment were 32.35% and 41.15%, respectively. Combined treatment with losartan slightly reduced the antitumor activity of axitinib to 21.50% and 40.48%, respectively. In addition, losartan alone showed strong antitumor activity with a TGI of 33.37%. The body weight changes of each group during the administration period are shown in Figure 5A. Figure 5B shows the effect of axitinib and axitinib plus losartan on xenograft tumor growth in BALB/c nude mice during 15 days of treatment. The tumor volumes following treatment with low-dose axitinib, high-dose axitinib, and losartan were 356.7, 310.2, and 349.1 mm^3^, respectively, which were significantly smaller than the volume in the model group. Axitinib inhibited tumor growth and this inhibition was positively correlated with drug concentration. Although the combined application of losartan did not significantly enhance the antitumor effect of axitinib, on the other hand, it did not affect the antitumor effect of axitinib.

#### 2.4.2. Effects of Axitinib and Axitinib Plus Losartan on Liver and Kidney Indices in Tumor-Bearing Mice

The effects of axitinib and axitinib plus losartan on liver and kidney indices were evaluated to determine the effects of these drugs on the liver and renal burden of tumor-bearing BALB/c nude mice. As shown in Figure 5C, low- and high-dose axitinib increased the kidney index (*p* < 0.05 for low-dose), whereas combined treatment with losartan slightly reduced this effect (*p* < 0.05 for low-dose axitinib plus losartan). As shown in Figure 5D, both low- and high-dose axitinib significantly increased the liver index (*p* < 0.05), whereas combined treatment with losartan reduced this effect. Losartan alone did not alter the liver and kidney indices of tumor-bearing mice.

#### 2.4.3. Effect of Axitinib and Axitinib Plus Losartan on Histopathological Changes in the Tumor Tissue

Tumor tissue from tumor-bearing mice was stained with hematoxylin and eosin to detect histopathological changes after drug administration. As shown in Figure 5F, tumor atypia and mitotic regions were clearly observed, with a delicate cytoplasm and small nucleus in the model group. In the treatment groups, the nucleus was larger and fewer than in the model, as well, mitotic regions were detected. In the low-dose axitinib group, the nuclei were large and transparent, and nucleoli were obvious. The nucleus in the high-dose axitinib group was more uniform and the cytoplasm volume was reduced. The nucleus in the losartan group was smaller than that in the axitinib group. In the low-dose axitinib plus losartan group, the large nucleoli of the nuclei were not obvious and cell proliferation was not active. In the high-dose axitinib plus losartan group, the nuclei were hyperchromatic and transparent. The results of hematoxylin–eosin staining further confirmed that the antitumor effect of axitinib was concentration-dependent, and the combined use of losartan would not affect the antitumor effect of axitinib.

## 3. Discussion

Although it has potent anticancer activity, long-term use of axitinib can cause severe side effects [10], particularly hypertension and proteinuria. One important factor in VEGF receptor inhibitor-related hypertension is downregulation of eNOs [20], with further decreased production of nitric oxide. Increased circulating endothelin-1, capillary rarefaction, and prostacyclin synthesis are also thought to be important mechanisms of hypertension associated with VEGF inhibition.

The mechanism of axitinib-related proteinuria is also related to the inhibition of VEGF pathways [21]. One study [22] showed that axitinib can induce proteinuria but has no severe negative effects on renal function, and the angiotensin receptor blocker drug candesartan alleviates axitinib-related proteinuria. In this study, we used another angiotensin receptor blocker drug, losartan, which has vascular protective effects and extensive anticancer activity in vivo and in vitro. The effects of losartan on axitinib-related high blood pressure, proteinuria, and increases in biochemical indices of liver and renal function were assessed. As a result, losartan significantly reduced axitinib-related high blood pressure increase and proteinuria, together with the levels of AST/ALT and creatinine in the rat serum, as well as the liver and renal indices in tumor-bearing mice. These results indicated that losartan can alleviate axitinib-related hypertension and liver and renal dysfunction. Furthermore, we focused on the inner mechanism of the interaction between losartan and axitinib-related endothelial dysfunction by measuring the levels of endothelin-1 and eNOs in rat serum. Endothelin-1 may contribute to VEGF receptor inhibitor-related hypertension. We found that combination treatment with losartan significantly reduced the axitinib-related blood pressure increase. One study [23] provided evidence that Ang II enhances ET-1-induced vasoconstriction through the elevation of ETAR expression and ET-1/ETAR binding. The relationship between the reduction of ET-1 and blood pressure and the interaction of Ang II and ET-1 are needed to elucidate. The level and activity of eNOs is thought to be another factor involved in VEGF receptor inhibitor-related hypertension and endothelin-1 dysfunction. Vascular endothelial cells can synthesize and release NO to relax smooth muscle and simultaneously release ET to contract smooth muscle, the imbalance between NO and ET produced by endothelial cells is a reason for the development of hypertension. As a downstream pathway of AKT/PI3K, inhibition of eNOs can have effects on vascular permeability and tumor growth. Treatment with axitinib for 7 days significantly reduced eNOs levels in rat serum. In addition, losartan treatment did not alleviate this effect; in fact, losartan showed similar effects on eNOs in the serum by reducing eNOs levels. The result of the increased blood pressure appeared to be a consequence of an impaired capacity for NO generation because of decreased NO synthase expression. The reduction of eNOs in the combination group requires further validation to explore whether it is related to angiotensin system inhibitors. But the data of heart rate and systolic blood pressure were obtained only for 7 days, the long-term impact of these drugs needs to be further studied, as well as further exploring the relationship between the expression level of eNOs and hypertension. The increased BP is not a result of RAS activation but rather appears to be a consequence of an impaired capacity for NO generation because of decreased NO synthase expression. Meanwhile, a Miles assay and xenograft tumor model were used to evaluate the effect of losartan on the anttumor activity of axitinib. Early work was done by Yanxia Zhao et al. [24] and described how losartan therapy alone did not reduce tumor burden, but it reduced both the incidence and the amount of ascites formed. L. XIAO et al. [25] thought treatment with losartan alone did not suppress tumor growth. But the research of Songtae Kim et al. [26] showed the survival of rats with orthotopic pancreatic cancer was significantly prolonged by treatment with either gemcitabine or losartan compared with that in the control group and that the survival was further prolonged by the combination of gemcitabine and losartan. In this study, losartan combined with axitinib showed strong antitumor activity in a xenograft tumor model of renal cell carcinoma, which has not been reported previously. The inhibition of vascular permeability is an important anticancer mechanism of VEGF receptor inhibitors. Losartan did not significantly affect axitinib-induced low skin vascular permeability or the anticancer activity of axitinib, but the TGI decrease in the group of combination of low-dose axitinib and losartan needs to be explored further. Taken together, axitinib plus losartan significantly alleviated hypertension, proteinuria, and changes to liver and renal function and did not significantly reduce the anticancer activity of axitinib. In addition to hypertension, and liver and kidney toxicity, fatigue and diarrhea are also other major issues of side effects in the treatment of renal cell carcinoma caused by axitinib. The follow-up will continue to evaluate the effect of combination therapy on side effects such as fatigue and diarrhea. This study lays the foundation for the wider clinical application of axitinib, but the precise mechanism of the interaction of the VEGF receptor inhibitor axitinib and angiotensin receptor blocker drugs requires further analysis.

## 4. Materials and Methods

### 4.1. Materials, Cell Culture, and Animals

Axitinib was synthesized by the Pharmaceutical Design and Synthesis Laboratory of Tianjin University of Science and Technology. Losartan was purchased from Anaiji Chemical Co., Ltd. (Shanghai, China). Trypsin 0.25% solution (Hyclone), RPMI (Roswell Park Memorial Institute) medium 1640 basic (Gibco) and fetal bovine serum were obtained from Solarbio Co., Ltd. (Beijing, China). PBS (Phosphate Buffer Saline, 10×) solution was prepared by NaCl (80.00 g), Na_2_HPO_4_•12H_2_O (29 g), KCl (2 g), KH_2_PO_4_ (2 g), which was dissolved in 1 L of sterile double distilled water. The formalin solution was made up of 100 mL of 40% formaldehyde solution and 900 mL of 1 × PBS solution. The aspartate aminotransferase assay kit, alanine aminotransferase assay kit, creatinine (Cr) assay kit (sarcosine oxidase), urine protein test kit, endothelin-1 assay kit, and endothelial nitric oxide synthase assay kit were purchased from Nanjing Jiancheng Bioengineering Institute. The 786-O (Human renal clear cell adenocarcinoma) was furnished from Peking Union Medical College. The Wistar rats, C57BL/6N mice, and BALB/c nude mice were purchased from Charles River Laboratories, Beijing, all animal experiments carried out in accordance with the National Institutes of Health guide for the care and use of Laboratory animals and approved by the Academic Committee of Tianjin University of Science and Technology.

### 4.2. Rat Blood Pressure Measurement

In this experiment, the tail-cuff method was used to detect the effect of axitinib and losartan as single agents and their combination on the systolic blood pressure of Wistar rats. Eight 6-week-old male Wister rats were used in the experiment. Axitinib was administered to mice at the dose of 25 mg/kg via intravenous injection. Losartan was administered to mice at the dose of 10 mg/kg by oral administration. Losartan in the combination group was taken orally 1 h after the administration of axitinib while control mice received physiological saline. The drug was administered twice a day, with an interval of 8 h for 7 consecutive days. Normal blood pressure and heart rate were measured for 3 days before administration. From the first day of administration, blood pressure and heart rate were measured every 4 h after the administration of the drugs and this day was designated as day 1. On the 6th day, the rats were placed in a metabolic cage, and 24 h urine was used for the determination of urine protein. On the 8th day, blood was taken from the femoral artery and the rats were sacrificed to take out the main organs used for liver and kidney function testing.

### 4.3. Liver and Renal Function

To assay the effects of axitinib and losartan as single agents and their combination on liver and kidney function in rats, the kit mentioned in the application method were used to detect the glutamate oxamate aminotransferase and glutamate alanine aminotransferase value in the serum and the content of urine protein in urine.

### 4.4. Skin Vascular Permeability (Miles Assay)

The Miles experiment was referenced and optimized to discuss the effects of drugs on skin vascular permeability. In this experimental study, 7-week-old female C57BL/6N mice were used. The experiment was divided into axitinib 10 mg/kg, losartan 10 mg/kg, axitinib + losartan (10 mg/kg + 10 mg/kg), and control group. After administration, 200 μL of piramine maleate was injected intraperitoneally to rule out the effect of endogenous histamine on vascular permeability. Evans blue dye was injected into the tail vein for 30 min. Mice were anesthetized with urethane. Then, VEGF-A (vascular endothelial growth factor-A) and 1 × PBS were injected intracutaneously on both sides. After that, the mice were returned to the cage for observation until the administration reached 4 h. Furthermore, the skin of each mouse was cut vertically from the lower abdomen to the chest about 3–4 cm. After drying the skin sample, formamide was added to extract the dye. Moreover, the liquid was heated by heater at 55 °C for 12 h. Finally, supernatant was taken after centrifugation to measure the absorbance at 620 nm on a spectrophotometer to test the skin vascular permeability.

### 4.5. Endothelin Function

To assay the effects of axitinib and losartan as single agents and their combination on the microvascular, an endothelin-1 assay kit and an endothelial nitric oxide synthase assay kit were used to assay the content of endothelin-1 (ET-1) and endothelial nitric oxide synthase (eNOs) in the serum in order to investigate the influence of the drugs on vascular physiology.

### 4.6. In Vivo Therapeutic Efficacy Studies

A xenograft tumor model was established by subcutaneous injection of the human kidney cancer cell 786-O to evaluate the inhibitory effect of axitinib and losartan and their combination on tumor growth. Treatments were initiated when tumors reached a volume of around 150 mm^3^. Axitinib was injected into mice at doses of 30/60 mg/kg, and losartan at doses of 10 mg/kg on day 0, 2, 4, 6, 8, 10, 12, and 14. The mice were euthanized on day 15. Tumor size and body weight were measured every 3 days. Serum and viscera of mice were extracted and stored for further research. Tumor pieces were excised and fixed in 10% formaldehyde and then embedded in paraffin. After that, tumor tissues were stained with hematorylin–eosin and examined under a light microscope. Dehydrated embedded sections and sealing sections were commissioned by Tianjin YiSheng Yuan Bio-technology Company (Tianjin, China).

### 4.7. Statistical Analysis

All the experiments were repeated at least three times. Data were expressed as mean ± standard deviation (S.D.). Results were analyzed by two-tailed student’s *t*-test for two groups and one-way analysis of variance for multiple groups. * *p* < 0.05, ** *p* < 0.01, *** *p* < 0.001 was considered statistically significant.

## Figures and Tables

**Figure 1 molecules-27-02764-f001:**
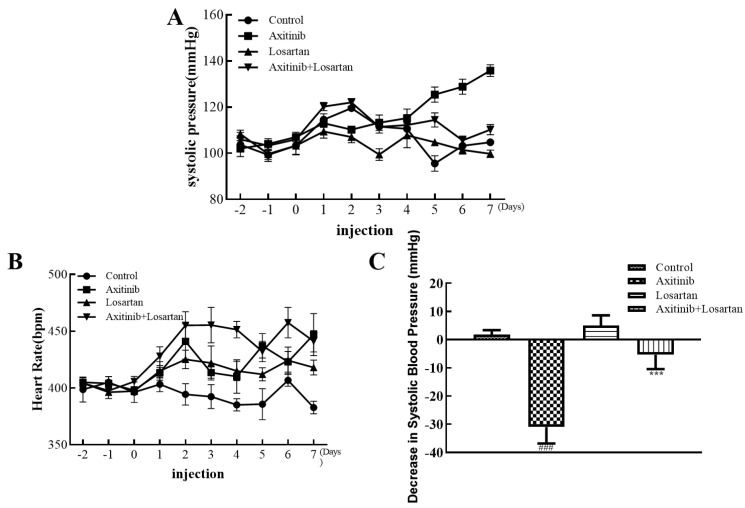
Effects of axitinib combined with losartan on systolic blood pressure and heart rate in rats. (**A**) Effects of axitinib combined with losartan on systolic blood pressure. (**B**) Effects of axitinib combined with losartan on heart rate. (**C**) The decreasing level of systolic blood pressure of rats in each group on the 7th day. ^###^
*p* < 0.001 vs. control. *** *p* < 0.001 vs. axitinib.

**Figure 2 molecules-27-02764-f002:**
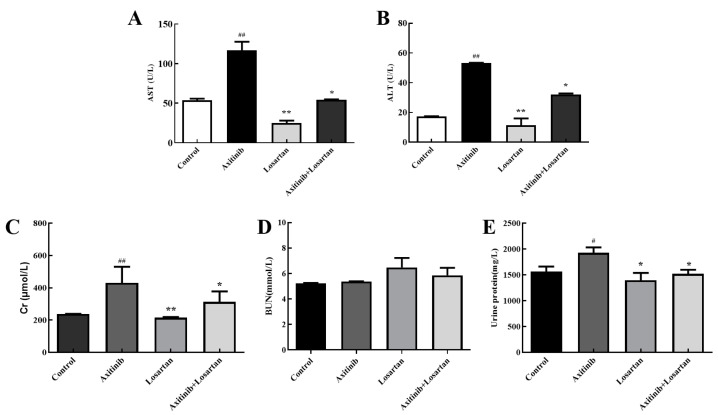
Effects of axitinib combined with losartan on the level of AST/ALT in rat serum and liver homogenate. (**A**) Effects of axitinib combined with losartan on the level of plasma AST in rats. (**B**) Effects of axitinib combined with losartan on the level of plasma ALT in rats. (**C**) Levels of plasma Cr in rats of experimental groups. (**D**) Effects of axitinib combined with losartan on the level of plasma BUN (blood urea nitrogen) in rats. (**E**) Levels of urine protein in rats of experimental groups. ^#^
*p* < 0.05 vs. control group, ^##^
*p* < 0.01 vs. control group, * *p* < 0.05 vs. axitinib group, ** *p* < 0.01 vs. axitinib group.

**Figure 3 molecules-27-02764-f003:**
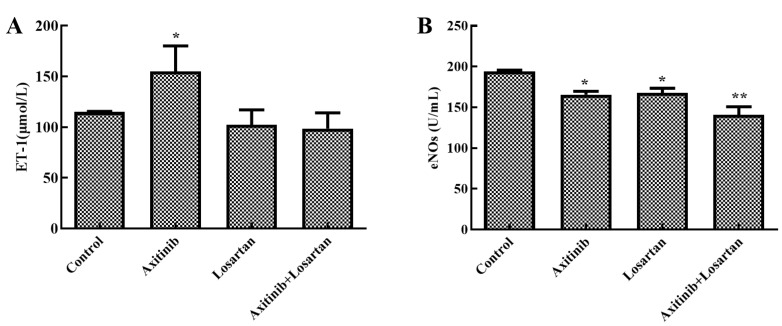
Effects of axitinib combined with losartan on endothelin function in rats. (**A**) Effects of axitinib combined with losartan on the level of endothelin-1 in rat serum. (**B**) Effects of axitinib combined with losartan on the level of eNOs in rat serum. * *p* < 0.05 vs. control. ** *p* < 0.01 vs. control.

**Figure 4 molecules-27-02764-f004:**
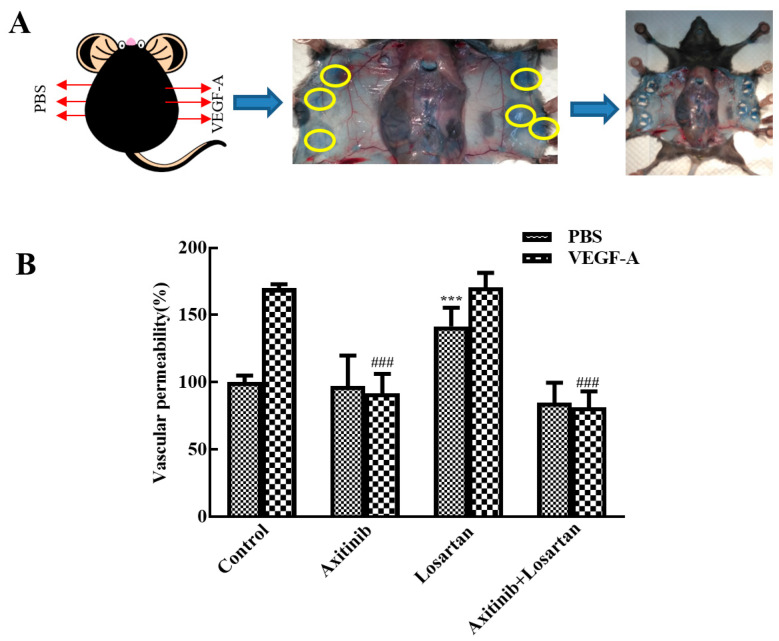
Effects of axitinib combined with losartan on skin vascular permeability in mice. (**A**) Experimental diagram. (**B**) Effect of different groups on skin vascular permeability. ^###^
*p* < 0.001 vs. VEGF-A group, *** *p* < 0.001 vs. control PBS group.

**Figure 5 molecules-27-02764-f005:**
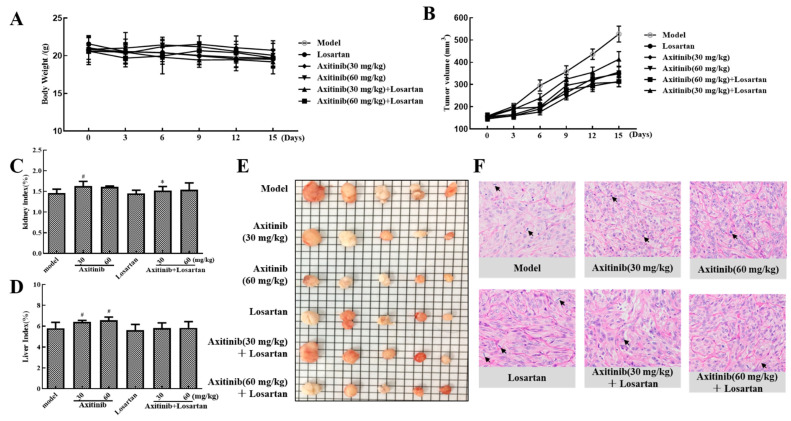
Effects of axitinib combined with losartan on xenograft tumor growth and organ index. (**A**) 15 days’ administration of axitinib combined with losartan on the body weight of mice. (**B**) 15 days’ administration of axitinib combined with losartan on the growth of xenograft tumor. (**C**) Effects of axitinib combined with losartan on kidney index of tumor-bearing mice. (**D**) Effects of axitinib combined with losartan on liver index of tumor-bearing mice. (**E**) Tumor pieces of each group after drug treatment. (**F**) Effects of axitinib combined with losartan on the histological changes of xenograft tumor. ^#^
*p* < 0.05 vs. model, * *p* < 0.05 vs. axitinib group.

**Table 1 molecules-27-02764-t001:** Effects of axitinib combined with losartan on the growth of xenograft tumor.

	Model	Axitinib Low Dose	Axitinib High Dose	Losartan	Axitinib-Low Dose + Losartan	Axitinib-High Dose + Losartan
Tumor volume/(mm^3^)	527.2	356.7	310.2	349.1	413.8	313.8
TGI/(%)	0.00	32.35	41.15	33.77	21.50	40.48

## Data Availability

The data presented in this study are available on request from the corresponding author.

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
