# Peer review of "Losartan Alleviates the Side Effects and Maintains the Anticancer Activity of Axitinib"

_molecules, 2022, doi:10.3390/molecules27092764_

Round 1

Reviewer 1 Report

In this study, the authors investigated the effects of Losartan on the side effects and antitumor activity of Axitinib by giving different drugs to various animal models. This study has some guiding significance for the use of antitumor drugs in clinic. At the same time, the method design of this study is unreasonable, and there are many problems in the results, so the results can not support the conclusions of this study. In addition, there are many mistakes in the whole article.

Major

  1. From line 292 to 294, “In this study, Losartan alone showed strong antitumor activity in a xenograft tumor model of renal cell carcinoma, which has not been reported previously.”However, it has been reported that RAS-blocking drugs including Losartan alone play a role in inhibiting renal cell carcinoma (PMID: 25595575, 9528828). It has even been studied in clinical trials that Angiotensin system inhibitors and survival in patients with metastatic renal cell carcinoma treated with VEGF-targeted therapy (PMID: 26685869, 21102591, 16101380).
  2. angiotensin-converting enzyme inhibitor
  3. When evaluating the effects of Axitinib and Losartan as a single drug and their combination on rats, the authors only applied the drugs to 8 rats divided into 4 groups, and observed and collected the relevant data for 7 days. We can see from Figure 1 that during these 7 days, the heart rate and systolic blood pressure of each group fluctuated greatly. Even sometimes the systolic blood pressure and heart rate of the combination group were higher. The authors only obtained data for seven days, but this could not allow us to further observe the long-term impact of the drug which is more concerned. Therefore, the authors need to observe the effect of Losartan combined with Axitinib on different tumor outcomes through longer animal experiments. It is also important that the number of rats is too small to make the data credible. In addition, the authors gave different drug doses to different model rats which need to be explained. Last but not least, published studies have shown that angiotensin-converting enzyme inhibitor could increase the incidence of lung cancer, so the authors need to further explain why losartan is used instead of other drugs.
  4. The main common side effects of Axitinib treatment for renal cell carcinoma include hypertension, fatigue, and diarrhea, etc.(PMID: 22056247, PMID: 19652060). But this study did not compare the influence of drugs on other common side effects among groups except hypertension.
  5. As can be seen from Figure 3B, the level of eNOs in the Axitinib plus Losartan group was the lowest value among all groups. The authors suggested that this further confirmed amelioration of side effect related to Axitinib by Losartan. On the contrary, previous studies have shown that VEGF plays an important role in blood pressure (PMID: 19652084, 22283728, 23625292, 25088996). Sincethe authors' concern about these mechanisms in hypertension caused by Axitinib, it is recommended to supplement the discussion of the effect of angiotensin system inhibitors on eNOS and ET-1.

Minor

  1. It is recognized that AST and ALT are markers of hepatocyte injury. It is not appropriate to detect the values of both in liver homogenate to judge liver damage.Line 118 to 119,the authors mentioned that, compared with Axitinib, the application of Losartan reduced the content of AST by 4.69 times. But what is shown in Figure 2C does not match this data. 
  2. Authors should carefully check and correct many mistakes in the manuscript.
  • From Figure 1C, it looks like systolic blood pressure went down in the Axitinib group and the combination group, which is obviously not consistent with the data.
  • In Figure 4B, the illustration is not accurate
  • In Figure 5B, the illustration is not accurate, it should be"liver index", not "kidney index", and from the picture, there is no significant difference in liver index between the combined group and the Axitinib group. In Figure 5C, the renal index of the combined group was significantly lower than that of the Axitinib group, but the statistically significant "*" sign was not marked in the Figure.
  • On page 2, lines 84-85, the authors’ interpretation of the results in Fig. 1B mentioned that the heart rate of the Axitinib group was the highest of all groups, but in the results presented in Fig. 1B, the heart rate of the Axitinib and Losartan group appeared to be the h
  1. On page 3, lines 128-132, the authors’ interpretation of the results in Fig. 2E did not mention that the Losartan group significantly increased creatinine compared to the Control group, which was marked in the figure.

Author Response

Dear reviewer,

Manuscript ID: molecules-1624370

TITLE: Losartan Alleviates the Side Effects and Maintains the Anticancer Activity of Axitinib

We are most grateful to you and the reviewers for the helpful comments on the original version of our manuscript entitled “Losartan Alleviates the Side Effects and Maintains the Anticancer Activity of Axitinib”. These comments are all valuable and very helpful for revising and improving my paper, as well as the important guiding significance to our researches. We have taken all comments into account and submit, herewith, a revised version of our paper. We marked newly added content in the paper as green and marked modified place as red.

We hope that the revised version of our paper is now suitable for publication in the Molecules and we look forward to hearing from you at your earliest convenience.

                                                                        Yours sincerely,

                                                                           Yuou Teng, Ph.D.

The responds to the reviewer’s comments are as follows:

In this study, the authors investigated the effects of Losartan on the side effects and antitumor activity of Axitinib by giving different drugs to various animal models. This study has some guiding significance for the use of antitumor drugs in clinic. At the same time, the method design of this study is unreasonable, and there are many problems in the results, so the results can not support the conclusions of this study. In addition, there are many mistakes in the whole article.

Major

  1. From line 292 to 294, “In this study, Losartan alone s howed strong antitumor activity in a xenograft tumor model of renal cell carcinoma, which has not been reported previously.”However, it has been reported that RAS-blocking drugs including Losartan alone play a role in inhibiting renal cell carcinoma (PMID: 25595575, 9528828). It has even been studied in clinical trials that Angiotensin system inhibitors and survival in patients with metastatic renal cell carcinoma treated with VEGF-targeted therapy (PMID: 26685869, 21102591, 16101380).
  • Thank you for your suggestion. We agree with you can’t anymore. The anti-tumor effect of Losartan was discussed earlier in the Discussion section. The description of the sentence “In this study, Losartan alone showed strong antitumor activity in a xenograft tumor model of renal cell carcinoma, which has not been reported previously” was our mistake and has been revised to “In this study, Losartan combined Axitininb showed strong antitumor activity in a xenograft tumor model of renal cell carcinoma, which has not been reported previously.”

  1. angiotensin-converting enzyme inhibitor
  • Thank you for your suggestion. Angiotensin System Inhibitor (ASI) was used in our research, which including angiotensin receptor blockers and angiotensin-converting enzyme inhibitors. We have checked the writing of the full text.

  1. When evaluating the effects of Axitinib and Losartan as a single drug and their combination on rats, the authors only applied the drugs to 8 rats divided into 4 groups, and observed and collected the relevant data for 7 days. We can see from Figure 1 that during these 7 days, the heart rate and systolic blood pressure of each group fluctuated greatly. Even sometimes the systolic blood pressure and heart rate of the combination group were higher. The authors only obtained data for seven days, but this could not allow us to further observe the long-term impact of the drug which is more concerned. Therefore, the authors need to observe the effect of Losartan combined with Axitinib on different tumor outcomes through longer animal experiments. It is also important that the number of rats is too small to make the data credible. In addition, the authors gave different drug doses to different model rats which need to be explained. Last but not least, published studies have shown that angiotensin-converting enzyme inhibitor could increase the incidence of lung cancer, so the authors need to further explain why losartan is used instead of other drugs.
  • Thank you for your advice. First, there are certain flaws in the design of the experiment of heart rate and systolic blood pressure. For this part of the experimental results, we have conducted an objective analysis in the discussion. Second, we selected different doses of Axitinib in this study based on the effects of Axitinib on tumor proliferation, regrowth of blood vessels, urinary protein excretion, renal function and hypertension in different literatures (PMID: 17016557, 26423685, 15215160, 19010843). The high antitumor dose of Axitinib selected in this study was doubled on the basis of the low dose (30 mg/kg). Third, the reasons for choosing Losartan have been added in the Introduction and marked as green.

  1. The main common side effects of Axitinib treatment for renal cell carcinoma include hypertension, fatigue, and diarrhea, etc.(PMID: 22056247, PMID: 19652060). But this study did not compare the influence of drugs on other common side effects among groups except hypertension.
  • Thank you for your suggestion. The progress of the experiment was limited to the effects of COVID-19. The limitations of this paper on the evaluation of side effects of the combination and next planning have been discussed.

  1. As can be seen from Figure 3B, the level of eNOs in the Axitinib plus Losartan group was the lowest value among all groups. The authors suggested that this further confirmed amelioration of side effect related to Axitinib by Losartan. On the contrary, previous studies have shown that VEGF plays an important role in blood pressure (PMID: 19652084, 22283728, 23625292, 25088996). Sincethe authors' concern about these mechanisms in hypertension caused by Axitinib, it is recommended to supplement the discussion of the effect of angiotensin system inhibitors on eNOS and ET-1.
  • Thank you for your reminder, your opinion can make the content of the article more comprehensive. The discussion on the effect of angiotensin system inhibitors on eNOS and ET-1 have been supplemented in the Discussion and marked as green.

Minor

  1. It is recognized that AST and ALT are markers of hepatocyte injury. It is not appropriate to detect the values of both in liver homogenate to judge liver damage.Line 118 to 119,the authors mentioned that, compared with Axitinib, the application of Losartan reduced the content of AST by 4.69 times. But what is shown in Figure 2C does not match this data.
  • Thank you for your comment. The data of losartan and combination groups in Figure 2 were confusing and the evaluation of liver injury in the literatures is mainly based on the content of ALT and AST in serum (PMID: 28576551, 35071372). The content of Figure 2 has been modified.

  1. Authors should carefully check and correct many mistakes in the manuscript.

From Figure 1C, it looks like systolic blood pressure went down in the Axitinib group and the combination group, which is obviously not consistent with the data.

  • Thank you for your advice. The data in Figure 1C means that the systolic blood pressure reduction level of each group on 7th day (PMID: 20733093), therefore, the group with an increase in the value is shown as a negative value in the figure. The legend in Figure 1C is unclear and has been changed.

In Figure 4B, the illustration is not accurate

  • Thank you for your advice. The diagram in Figure 4 lacks self-explanatory and has been revised.

In Figure 5B, the illustration is not accurate, it should be"liver index", not "kidney index", and from the picture, there is no significant difference in liver index between the combined group and the Axitinib group. In Figure 5C, the renal index of the combined group was significantly lower than that of the Axitinib group, but the statistically significant "*" sign was not marked in the Figure.

  • Thank you for your suggestion. We have changed and checked the data, thank you again for your carefulness.

On page 2, lines 84-85, the authors’ interpretation of the results in Fig. 1B mentioned that the heart rate of the Axitinib group was the highest of all groups, but in the results presented in Fig. 1B, the heart rate of the Axitinib and Losartan group appeared to be the h highest.

  • We are so sorry for the clarity of Fig. 1B, because the clarity of the picture is not enough, the representative symbol of the intersection of the curves is not clear. The figure has been replaced.

  1. On page 3, lines 128-132, the authors’ interpretation of the results in Fig. 2E did not mention that the Losartan group significantly increased creatinine compared to the Control group, which was marked in the figure.
  • Thank you for your suggestion. We have changed the format of statistical analysis in a proper and accurate format. The interpretation of the results in Fig. 2 has added and marked in green.

Reviewer 2 Report

General Comments:

This manuscript presented an interesting approach to reduce side effects of axitinib which is one of the most potent inhibitors of vascular endothelial growth factor (VEGF) receptor with strong antitumor activity toward various malignant tumors (renal carcinoma, lung cancer, colorectal cancer, breast cancer, melanoma and thyroid cancer). To explore the effects of Losartan which is angiotensin receptor blocker on side effects caused by axitinib and its antitumor activity several animal experiments were conducted. The results showed that the systolic blood pressure of Wistar rats was significantly increased by 30 mmHg in 7 days of Axitinib treatment, while combination of Losatran significantly reduced the blood pressure rise caused by Axitinib. The results of Miles experimental model and mouse xenograft tumor model demonstrated that Losartan has no significant influence on Axitinib related low vascular permeability and antitumor activity. The manuscript showed that combination of Axitinib and Losartan significantly reduced the side effects and maintained the antitumor effects of Axitinib and provides an idea for overcoming VEGF receptor inhibitor-related side effects. This topic should be of interest for the community of medicine and pharmaceutical science. Therefore, I suggest an acceptance for publication of this manuscript. 

Author Response

Thank you for your comments and acknowledgment of the subject.

Reviewer 3 Report

In the present study, an Axitinib-Losartan combination strategy was established in order to suppress Axitinib-related side effects. The effects of Losartan on Axitinib-related high blood pressure, proteinuria, and increases in biochemical indices of liver and renal function, such as AST/ALT,  creatinine, endothelin-1 and eNOs, were assessed. The Miles experimental model and mouse xenograft tumor model were further used to evaluate the effect of Losartan on the anti-tumor effect of Axitinib. These results indicate that Axitinib plus Losartan significantly alleviated hypertension, proteinuria, and changes to liver and renal function and did not significantly reduce the anticancer activity of Axitinib.

However, this work needs minor revisions before publication in Molecules. There are several questions as follow:

  1. Statistical analysis should be present in a proper and accurate format. It would be better to make it easier for readers to understand. Moreover, correct symbol should be used.
  2. In lines 206, ***p<0.001 vs PBS group should be corrected into ***p<0.001 vs control PBS group.
  3. In lines 84-86, The result description in Fig. 1B is not accurate enough, and the result description does not accord with the curve in Fig. 1B.
  4. In Figure 2B and 2E, Axitinib and Losartan alone increased the level of ALT in plasma and liver homogenates, but the level of ALT decreased after Axitinib-Losartan combination were used. Please explain the reasons.
  5. In lines 210-215, the tumor growth inhibition (TGI) ratio of low-dose Axitinib and Losartan alone was 32.35% and 33.37% respectively. However, the TGI decreased significantly in the group of combination of low-dose Axitinib and Losartan. Please make a proper explanation.
  6. The clarity of the full-text picture is not enough, and the color contrast of different groups in the statistical map is poor.
  7. For in vivo assays, the data of body weight curve should be supplemented.

Round 2

Reviewer 1 Report

  1. As mentioned above, there are major defects in the experimental part of measuring blood pressure and heart rate of mice. This part of the experiment should be improved as much as possible, since it is extremely important for the development of subsequent experiments. In addition, the authors mentioned in the reply that there was an analysis in the discussion section, but it was not reflected in the manuscript.
  2. In the y-axis label of Figure 1C, there was a grammatical error in "Dipping System Blood Pressure", and it is suggested to use "Decrease in Systolic Blood Pressure"
  3. In line 175, the interpretation of urinary protein results should quote Figure 2E instead of Figure 2D.
  4. The annotation of statistically significance was not consistent in Figures 3A and 3B, which would cause confusion. It is suggested to uniform labeling. It is suggested to annotate in a consistent way.
  5. The authors have modified the arrangement of Figure 5, however there is no corresponding modification in the interpretation in line 279-320, and causes inconsistencies. Please correct the of the interpretation of the content of Figure 5 in Results section.
  6. According to the interpretation of the results in line 296-303, it seems that Figure 5C is the result of the liver index, and Figure 5D is the result of the kidney index, which is inconsistent with the labels in the figure. Please explain it further.
  7. In lines 238-240, the authors noted that the eNOs evaluation also confirmed that Losartan improved the associated side effects of Axitinib. However, in the Line 372-374, the sentence “Result of the increased blood pressure appeared to be a consequence of an impaired capacity for NO generation because of decreased NO synthase expression.”, and the level of eNOS is the lowest in the Axitinib plus Losartan group. The data suggested that eNOs was the lowest in the Losartan plus Axitinib group. the authors also mentioned that the increase of BP was not the result of RAS activation, but the result of NO production ability impaired by the decreased expression of NO synthase. This is inconsistent and makes people confused about the role of eNOs in blood pressure changes in each group. How does a reduction in eNOS improve blood pressure in the combined group? Further elaboration is recommended.
